# Metabolic Syndrome Ameliorated by 4-Methylesculetin by Reducing Hepatic Lipid Accumulation

**DOI:** 10.3390/ijms231810465

**Published:** 2022-09-09

**Authors:** Linghuan Li, Guangyao Zhu, Gaohang Fu, Weiwei Zha, Hanbing Li

**Affiliations:** 1Institute of Pharmacology, Zhejiang University of Technology, Hangzhou 310014, China; 2Section of Endocrinology, School of Medicine, Yale University, New Haven, CT 06520, USA

**Keywords:** visceral obesity, hepatic steatosis, 4-methylesculetin, adipose microenvironment, metabolic syndrome

## Abstract

Obesity is a chronic metabolic disease caused by an imbalance between energy intake and expenditure during a long period and is characterized by adipose tissue disfunction and hepatic steatosis. The aim of this study was to investigate the effect of 4-methylesculetin (4-ME), a coumarin derivative, upon adipose microenvironment and hepatic steatosis in mice induced by a high-fat diet (HFD), and to explore potential mechanisms of its beneficial effect on metabolic disorders. HFD-fed mice displayed visceral obesity, insulin resistance, and hepatic lipid accumulation, which was remarkably ameliorated by 4-ME treatment. Meanwhile, 4-ME ameliorated adipocyte hypertrophy, macrophage infiltration, hypoxia, and fibrosis in epididymal adipose tissue, thus improving the adipose tissue microenvironment. Furthermore, 4-ME reversed the increase in CD36, PPAR-γ, SREBP-1, and FASN, and the decrease in CPT-1A, PPAR-α, and Nrf2 translocation into the nucleus in livers of HFD mice and in FFA-incubated hepatocytes. Moreover, the beneficial effects of 4-ME upon lipid deposition and the expression of proteins related to lipid metabolism in FFA-induced LO2 cells were abolished by ML385, a specific Nrf2 inhibitor, indicating that Nrf2 is necessary for 4-ME to reduce hepatic lipid deposition. These findings suggested that 4-ME might be a potential lead compound candidate for preventing obesity and MAFLD.

## 1. Introduction

Obesity refers to excessive fat accumulation and ectopic body mass increase, which is a multi-factor chronic metabolic disease [1]. It is not only the proliferation and hypertrophy of adipocytes, but also the apoptosis and necrosis of adipocytes and preadipocytes. Obesity can cause many metabolic diseases, such as diabetes, cardiovascular diseases, fatty liver, and cancer, and a common characteristic of insulin resistance. According to World Health Organization reports, worldwide obesity has nearly tripled since 1975. In 2016, more than 1.9 billion adults 18 years and older were overweight. Of these, over 650 million were obese. Therefore, obesity is a critical public health problem worldwide [2,3].

With deeper knowledge of adipose tissue, it is no longer regarded as an inert energy storage organ. The discovery of leptin and other important adipose-derived hormones makes the new definition of adipose tissue as an endocrine organ, which has been extensively accepted [4]. The abnormality of adipose tissue and its microenvironment is of great significance to the metabolism of other tissues and organs as well as the whole body [5]. Commonly, the fat microenvironment in obese patients changes, causing macrophage infiltration [6]. At the same time, excessive accumulation of fat can lead to inflammation and disorder of adipokine secretion, chronic inflammation, and a hypoxic microenvironment, and ultimately leads to the production of adipose fibrosis [7]. In addition, hypoxia can increase the expression of hypoxia-inducible factor 1α (HIF1α) in adipose tissue, activate the signal pathway related to Transforming Growth Factor β (TGF-β), and promote the infiltration of immune cells, leading to adipose fibrosis [8]. The interaction of hypoxia, inflammation, and fibrosis in the adipose tissue microenvironment plunges the microenvironment into a vicious circle [9]. When obesity is induced by internal and external factors such as high fat, the microenvironment of adipose tissue deteriorates, which usually causes or aggravates the accumulation of fat in other tissues (such as liver, muscle, etc.), leading to diabetes, fatty liver, and so forth. In obese patients, plasma non-esterified fatty acid (NEFA) levels are usually elevated because of the enhancement of adipose tissue lipolysis. Roughly 60% of triglyceride (TG) arises from the plasma NEFA pool, 30% from de novo lipogenesis, and 10% from the diet in the liver of obese individuals [10]. Furthermore, adipose tissue failure or dysfunction may signal progression of hepatic steatosis toward NASH [11]. Thus, the adipose tissue microenvironment could be targeted for the treatment of obesity and metabolic-associated fatty liver disease (MAFLD).

The coumarin derivative 4-methylesculetin (6,7-dihydroxy-4-methylcoumarin, 4-ME) is isolated from *Artemisia annua* [12]. Coumarins and their derivatives have drawn much attention due to a wide range of bioactivities such as antioxidant [13], antiarthritic, anti-inflammatory properties [14,15,16,17], and beneficial effects upon obesity and its related comorbidities as well [18,19,20,21,22]. However, whether 4-ME ameliorates adipose tissue dysfunction and obesity-related insulin resistance remains to be elucidated. The aim of this study was to investigate the effect of 4-ME on the adipose tissue microenvironment and hepatic steatosis induced by a high-fat diet, and to explore potential mechanisms of its beneficial effect upon metabolic disorders, so as to lay a foundation for further research and development.

## 2. Results

### 2.1. Effects of 4-ME upon Visceral Obesity and Insulin Resistance in HFD-Fed Mice

To investigate the effects of 4-ME upon obesity and its related metabolic disorders, mice were fed an HFD for 8 weeks and then treated with 15 mg/kg 4-ME, 50 mg/kg 4-ME, or 250 mg/kg metformin. As shown in Table 1, the average body weight was significantly lower in the 50 mg/kg 4-ME group and the metformin group compared with the Model group after the 8-week treatment, as well as epididymal adipose tissue weight manifested by the ratio of epididymal fat/body weight and the Lee Index defined by Lee in 1929 as the cube root of body weight (g) divided by the naso-anal length (cm) [23]. In addition, 4-ME and metformin did not cause changes in food intake (Appendix A).

Meanwhile, serum biochemical parameters in mice were determined to assess the effects of 4-ME upon obesity and its related disorders induced by HFD. Mice fed an HFD had increases in serum levels of FBG, insulin, and leptin compared with the Control group, which were reversed by 4-ME, especially 50 mg/kg 4-ME, and metformin (Table 2). Insulin resistance shown as HOMA-IR induced by HFD was significantly ameliorated by 4-ME and metformin. The results are consistent with those of the intraperitoneal glucose tolerance test (IPGTT), which show that the impaired glucose tolerance induced by HFD was improved by 4-ME and metformin (Figure 1A,B).

### 2.2. Effects of 4-ME upon Adipose Tissue Microenvironment in HFD-Fed Mice

The adipose tissue hypertrophy and inflammation in HFD-fed mice were investigated by examining adipocyte size and macrophage infiltration in epididymal adipose tissue sections stained with hematoxylin and eosin. Masson’s trichrome staining was introduced to measure collagen levels. The hypoxia and fibrosis in epididymal adipose tissue were detected via immunohistochemistry using HIF1α and TGF-β antibodies. As shown in Figure 1C–F, the increased adipocyte size, macrophage infiltration, upregulated collagen level, and HIF1α and TGF-β expression in the epididymal adipose tissue in HFD-fed mice were ameliorated by 4-ME and metformin, indicating the improvement of the adipose tissue hypertrophy, inflammation, hypoxia, and fibrosis.

### 2.3. Effects of 4-ME upon Lipid Metabolism and Hepatic Steatosis in HFD-Fed Mice

To examine whether 4-ME regulates lipid metabolism, the serum levels of TG, NEFA, T-CHO, LDL-C, and HDL-C were measured in mice fed an HFD for 16 weeks. The HFD increased serum levels of TG, NEFA, T-CHO, and LDL-C compared with the Control group, while 4-ME and metformin decreased all these serum parameters, but only NEFA with statistical significance. In contrast, the HFD downregulated the serum level of HDL-C, while 4-ME and metformin upregulated it, but only metformin had a significant difference (Table 2). These results indicate that 4-ME can alleviate obesity-associated serum lipid metabolic disorders.

Then, the liver tissue sections were stained with H&E and Oil red O, respectively (Figure 2A). Histological observations in both H&E and Oil red O staining show that visible hepatic lipid accumulation in HFD-fed mice was remarkably reduced by 4-ME and metformin. In addition, the HFD-induced increase in hepatic TG accumulation was significantly decreased by high-dose 4-ME treatment (Figure 2B), suggesting that 4-ME prevents hepatic steatosis induced by an HFD.

### 2.4. Effects of 4-ME on the Expression of Proteins Involved in NEFA Uptake, Lipogenesis, and Fatty Acid β-Oxidation in Livers of HFD-Fed Mice

Furthermore, the expression of proteins related to NEFA uptake, lipogenesis, and fatty acid β-oxidation was analyzed in liver samples of HFD-fed mice. We found that the HFD increased the protein expression of cluster of differentiation 36 (CD36), peroxisome proliferator-activated receptor γ (PPAR-γ), sterol regulatory element-binding protein 1c (SREBP-1c), and fatty acid synthase (FASN), and decreased the expression of peroxisome proliferator-activated receptor α (PPAR-α) and carnitine palmitoyltransferase 1A (CPT-1A) compared with the Control group. Treatment with 4-ME counteracted the expression changes of these proteins, indicating the beneficial effects of 4-ME on lipid metabolism in the livers of HFD-fed mice (Figure 2C–H).

### 2.5. Effects of 4-ME upon Lipid Deposition in FFA-Incubated LO2 Cells

As shown in Figure 3A, treatment of LO2 cells with different concentrations (1, 3, 10, 30, and 100 μM) of 4-ME had no significant influence on cell viability. Then, the effects of 4-ME upon lipid metabolism in free fatty acid (FFA)-incubated LO2 cells were evaluated using Oil red O staining. In Oil red O staining (Figure 3B), the intracellular lipid droplets were stained red. After the 4-ME treatment, the intracellular lipid droplets decreased in a dose-dependent manner (1, 3, 10, 30, and 100 μM). In addition, metformin inhibited lipid accumulation compared with the FFA group. In addition, 4-ME treatment reduced the content of intracellular TG (Figure 3C). Taken together, these findings uncover that 4-ME treatment can ameliorate the accumulation of intracellular lipid droplets in FFA-incubated LO2 cells.

We next explored the effects of 4-ME on the expression of genes and proteins related to lipid metabolism. We found that the FFA challenge significantly increased the mRNA levels of CD36, FASN, SREBP1, and PPARG in LO2 cells compared with the Control group, whereas these alternations were dramatically abrogated when supplemented with different concentrations of 4-ME (Figure 3D–G). In addition, FFA incubation displayed the downregulation of the mRNA expression levels of PPARA and CPT1A, but 4-ME prevented the reduction in the mRNA expression of these genes (Figure 3H,I). Consistently, the results of Western blotting presented that an enhancement in protein expression of CD36, SREBP-1c, FASN, and PPAR-γ was reversed by 4-ME administration, and a reduction in protein expression of PPAR-α and CPT-1A was rescued by 4-ME treatment in FFA-incubated LO2 cells (Figure 4). Collectively, these data reveal that 4-ME can ameliorate lipid accumulation through regulating fatty acid uptake, β-oxidation, and lipogenesis in FFA-induced hepatocytes.

### 2.6. Nrf2 Is Necessary for 4-ME Reducing Lipid Deposition

Nuclear factor erythroid 2-related factor 2 (Nrf2) is a transcriptional factor that not only regulates a battery of cellular defense elements against oxidative stresses, but also acts as a regulator of cellular lipid configuration in the liver [24]. Western blotting results showed that in the liver of HFD-fed mice, Nrf2 translocation into the nucleus was decreased, while it was enhanced by 4-ME treatment. Meanwhile, in FFA-induced LO2 cells, the protein levels of nuclear Nrf2 were reduced, whereas they were significantly increased by 4-ME in a dose-dependent manner (Figure 5A–D).

To further elucidate the role of Nrf2 in lipid metabolism, a specific Nrf2 inhibitor, ML385, was used to treat FFA-induced LO2 cells prior to 4-ME incubation. The beneficial effects of 4-ME upon lipid deposition and the expression of proteins related to lipid metabolism in FFA-induced LO2 cells were abolished by ML385 (Figure 5E–K), indicating that Nrf2 acts as a negative regulator participating in hepatic lipid metabolism, which is consistent with previous reports [25].

## 3. Discussion

Obesity is characterized by lipid metabolic disorder. In this study, mice fed an HFD for 16 weeks displayed an increase in body weight, epididymal fat mass, epididymal adipocyte size, and the Lee Index, indicating visceral obesity that is a marker of ectopic fat accumulation in several key organs [26] and a critical mediator of steatohepatitis in metabolic liver disease [11]. Among these, intrahepatic fat accumulation is the hallmark profile of MAFLD [27]. The HFD-fed mice showed visible lipid droplets in the liver and elevated intrahepatic triglyceride accumulation, which was remarkably reduced by treatment with 4-ME and metformin.

Adipose tissue plays an important role in lipid metabolism. Adipose tissue dysfunction is characterized by disorders of the adipose tissue microenvironment including adipose hypertrophy, macrophage infiltration, hypoxia, fibrosis, and insulin resistance. It is extensively known to play a critical role in the pathogenesis of metabolic disorders such as MAFLD [28]. Adipocyte hypertrophy leads to adipocyte dysfunction and insulin resistance and induces local adipose tissue hypoxia manifested by activation of a critical transcriptional factor HIF1α that can accelerate adipose tissue fibrosis [29,30]. Adipose tissue fibrosis is characterized by the deposition of excessive extracellular matrix (ECM), mainly collagens, which can be promoted by a potent profibrotic factor TGF-β and lead to adipose tissue dysfunction and ultimately metabolic complications [9]. Furthermore, in hypertrophic adipocytes, elevated lipolysis leads to the release of large amounts of NEFAs, which can be taken by other tissues. Excessive NEFA uptake by the liver can induce increased lipid synthesis and gluconeogenesis. Increased serum levels of NEFAs can lead to peripheral insulin resistance [31]. In our study, the phenomena of adipocyte hypertrophy, macrophage infiltration, hypoxia, and fibrosis in obese mice were observed by means of histopathology and immunohistochemistry. Metformin and 4-ME could significantly ameliorate the macrophage infiltration, collagen level, and HIF1α and TGF-β expression in the epididymal adipose tissue in HFD-fed mice, indicating the improvement of the epididymal adipose tissue microenvironment, thus counteracting the disorder of lipid metabolism in the liver caused by obesity.

Additionally, hepatic lipid accumulation results either from increased NEFA uptake and de novo lipogenesis in the liver and/or decreased fatty acid β-oxidation [32]. NEFA uptake is facilitated by cell-surface receptor CD36 whose expression is normally low, while much higher in response to an HFD [33]. FASN is a key enzyme in the de novo lipogenesis pathway that is responsible for the synthesis of excess fat in the liver of patients with MAFLD and is regulated by SREBP-1 and PPAR-γ [34]. Furthermore, Nrf2 is considered as a negative regulator of hepatic lipid metabolism, which not only participates in antioxidant pathways, but also can indirectly suppress the expressions of SREBP-1 and its lipogenic target genes [25,35,36]. CPT-1A is essential for fatty acid oxidation, a process that metabolizes fats and converts them into energy, which is regulated by transcriptional factor PPAR-α [37]. In our study, the HFD-treated mice or FFA-incubated hepatocytes displayed increased protein expression of CD36, SREBP-1, PPAR-γ, and FASN, decreased protein expression of PPAR-α and CPT-1A, and reduced nuclear translocation of Nrf2 as well, leading to increased FFA uptake, elevated de novo lipogenesis, and suppressed fatty acid oxidation, while 4-ME dose-dependently reversed these changes, contributing to the beneficial effects of 4-ME upon lipid metabolism both in vivo and in vitro. Further experiment results show that Nrf2 is required for 4-ME to ameliorate lipid deposition in livers of HFD mice and in FFAs-incubated hepatocytes. In addition, we found that 4-ME alleviated lipid accumulation by Nrf2 activation in hepatocytes, but the mechanism for the 4-ME effect in the adipose tissue is yet to be elucidated. Several studies reported controversial roles of Nrf2 in the adipogenesis in adipose tissue depending on the different approaches and animal models [38,39]. Therefore, more research concerning the tissue-specific effects of Nrf2 KO and Nrf2 overexpression and thereafter the mechanism of 4-ME in improving the adipose tissue microenvironment requires further investigation.

Taken together, 4-ME attenuated adipose tissue dysfunction and hepatic steatosis, and thus improved metabolic syndrome (Figure 6), providing a potential lead compound candidate against obesity and MAFLD. Meanwhile, esculetin has been reported to exert beneficial effects upon liver fat accumulation [40,41]. Given the structural similarity of these two coumarin compounds, it is worthy of further investigation in the future, especially with head-to-head comparative experiments, in order to elucidate the structure–activity relationship and provide clues for drug development. From the pharmacokinetic aspect, Li et al. [42] investigated the bioavailability and corresponding mechanisms of these two compounds and reported that the average absolute bioavailability of esculetin and 4-ME was 10.07% and 22.28%, respectively, indicating that 4-ME has a higher bioavailability than esculetin. However, the systematic investigation of in vivo biotransformation of coumarins is warranted to see if the interconversion between esculetin and 4-ME exists. In further studies, we will conduct in-depth research on the pharmacokinetics of 4-ME and the mechanisms and the precise target of 4-ME.

## 4. Materials and Methods

### 4.1. Materials

The high-fat diet (HFD) consists of 15% lard (*w*/*w*) and 15% sucrose (*w*/*w*), 70% basic diet which was obtained from Zhejiang Academy of Medical Sciences (Hangzhou, China). The ingredients of the high-fat diet and standard chow diet are shown in Appendix A. The 4-ME was obtained from Chengdu Biopurify Phytochemicals Ltd. (Chengdu, China) with a purity of 99.95% in Appendix A. Insulin and leptin Elisa kits were provided by Shanghai Hengyuan Biological Technology Co., Ltd. (Shanghai, China); ML385 was purchased from Glpbio (Montclair, NJ, USA); metformin (purity more than 98%) and fatty free-bovine serum albumin (BSA) were obtained from Sangon Biotech (Shanghai, China). The following items were purchased from the cited commercial sources: CD36, CPT-1A, FASN, PPAR-α, PPAR-γ, TGF-β, Tubulin, and Lamin B antibodies (Proteintech, Chicago, IL, USA); HIF1α antibody (Invitrogen, Carlsbad, CA, USA); SREBP-1c antibody (Novus Biologicals, Littleton, CO, USA); Nrf2 antibody (Affinity Biologicals, Shanghai, China).

### 4.2. Animals Studies

Eight-week-old male wild-type ICR mice were purchased from Zhejiang Academy of Medical Sciences (Hangzhou, China) (License Number: SCXK (Zhe) 2014-0001) and housed under a temperature-controlled (23 ± 2 °C) and humidity-controlled (50–60%) environment with a light–dark cycle (12 h, respectively). After one week of acclimatization, model mice were fed with an HFD ad libitum while Control mice were fed with a low-fat diet, also ad libitum. Eight weeks later, model mice were divided into four groups, namely Model, 15 mg/kg 4-ME, 50 mg/kg 4-ME, and metformin, each group consisting of 8 mice. The Control and Model animals were gavaged with an equivalent volume of 0.5% sodium carboxymethyl cellulose (CMCNa_2_) solution; 15 mg/kg 4-ME group: 4-methylesculetin (15 mg/kg/day, p.o.); 50 mg/kg 4-ME group: 4-methylesculetin (50 mg/kg/day, p.o.); and metformin group: metformin (250 mg/kg/day, p.o.), respectively, for 8 weeks. According to the EPA TSCA Section 8(b) chemical inventory data, the lethal dose of 4-ME when administered orally is above 3000 mg/kg body weight for rodents (RTECS No.: GN6384500), and the high dose of 4-ME was selected according to the study of Hemshekhar et al. [15] Body weights were measured bi-weekly. After fasting on the last day of the study, all animals were euthanized humanely for serum and tissue analysis. The serum was separated and stored at −80 °C. Tissues were harvested, weighed, snap-frozen in liquid nitrogen, and stored at −80 °C. All animal operations were conducted in compliance with the guidelines for animal care and use of the Zhejiang University of Technology Laboratory Animal Center.

### 4.3. Cell Culture and Treatment

LO2 (HL-7702) cells, a human normal liver cell line, were provided by Procell Life Science & Technology Co., Ltd. (Wuhan, China). Cell culture and FFAs-BSA mixture (containing 0.33 mM oleic acid and 0.17 mM palmitic acid) were the same as described previously [43]. LO2 cells were divided into 8 groups, the Control group, FFAs group, FFAs+1 μM 4-ME group, FFAs+3 μM 4-ME group, FFAs+10 μM 4-ME group, FFAs+30 μM 4-ME group, FFAs+100 μM 4-ME group, and metformin group; each group was given the corresponding treatments. ML385 (10 μM), a specific Nrf2 inhibitor, was used to treat LO2 cells. The cells were incubated with freshly prepared medium supplemented with FFAs-BSA mixture for 24 h to induce hepatocyte steatosis and concurrently added with different concentrations of 4-ME or ML385.

### 4.4. Cell Viability Assay

The MTT method was adopted to assess cell viability. Briefly, LO2 cells were treated with different concentrations of 4-ME (1, 3, 10, 30, and 100 μM) in a 96-well plate. After 24 h of incubation, 20 μL of MTT solution (5 mg/mL) was added into each well. Following incubation for 4 h, the solution was replaced by 150 μL dimethyl sulfoxide to dissolve the formazan crystal. Finally, the optical densities were measured at 490 nm on a microplate reader (BioTek Synergy H1, Winooski, VT, USA).

### 4.5. Hepatic and Intracellular Triglyceride Determination

Evaluation of triglyceride content was achieved using a kit (Jiancheng Bioengineering Institute, Nanjing, China), according to the manufacturer’s instructions. Briefly, 100 mg of liver tissue was homogenized in 1 mL ethanol and then centrifuged at 2500× *g* for 10 min. Hepatic TG levels were measured with TG Quantification Kits. The cell samples were collected from 6 cm dishes to determine intracellular TG levels by TG Quantification Kits. The protein concentration of lysates was determined by Bicinchoninic Acid protein assay kit with BSA as a standard (Beyotime, Haimen, China).

### 4.6. Oil Red O Staining

After fixation, liver specimens were embedded in OCT and stored at −80 °C. Then, frozen liver sections (6 μm thick) were stained with Oil red O solution for 15 min at room temperature and stained with hematoxylin. Eventually, all sections were observed by microscopy. The cell samples were fixed with paraformaldehyde for 20 min, then stained with Oil red O for 15 min at room temperature and stained with hematoxylin. Images were obtained by microscopy.

### 4.7. Intraperitoneal Glucose Tolerance Test

All animals strictly fasted overnight. The mice were injected intraperitoneally with glucose (2 g/kg body weight). Blood samples were collected from the tail tips of mice at 0, 15, 30, 60, and 120 min, and blood glucose was measured using a glucometer.

### 4.8. Serum Analysis

Serum levels of T-CHO, LDL-C, HDL-C, NEFA and TG, and insulin were measured with kits from Nanjing Jiancheng Bioengineering Institute (Nanjing, China). Fasting serum insulin and leptin were analyzed by Elisa kits.

### 4.9. Histological and Immunohistochemistry Studies

Tissue specimens were fixed in 4% paraformaldehyde solution, embedded in paraffin, stained with hematoxylin and eosin (H&E), and Masson’s trichrome, respectively, after tissue dehydration with gradient ethanol, and assessed for histopathological changes using image analysis system. Adipocyte size was determined in hematoxylin/eosin-stained sections as the mean cell area (in μm^2^) of 300 random adipocytes on digital images acquired at 100× magnification by light microscope (Nikon) using IPP software (Image-Pro Plus 6.0). Adipose hypoxia and fibrosis were also detected using HIF1α and TGF-β antibodies. Adipose tissue sections were incubated with HIF1α (1:100) and TGF-β (1:100) primary antibodies, respectively, and a secondary antibody and then diaminobenzidine as a color substrate. At 200× magnification, the area of the staining signal (HIF1α and TGF-β) was measured using the IPP software.

### 4.10. Western Blotting

Tissue specimens were homogenized in ice-cold lysis buffer to extract the protein. Lysates were centrifuged at 14,000× *g* for 5 min at 4 °C and supernatants were collected. The protein concentration was determined by Bicinchoninic Acid protein assay kit with BSA as a standard (Beyotime, Haimen, China). Protein samples were mixed with loading buffer and boiled for 5 min, then resolved by SDS-PAGE and transferred onto 0.22 μm PVDF membranes (Bio-Rad, Hercules, CA, USA) in a semidry transfer system (Bio-Rad, Hercules, CA, USA). PVDF membranes were blocked with freshly prepared 5% BSA or non-fat milk in Tris-buffered saline containing 0.05% Tween-20 surfactant for 60 min at room temperature, immunoblotted with the primary antibodies for 4 h at room temperature or overnight at 4 °C, washed with TBST 3 times and incubated with secondary antibodies for 2 h at room temperature, and finally visualized with enhanced chemiluminescence method.

### 4.11. Total RNA Extraction and Quantitative RT-PCR

Total RNA was extracted from LO2 cells using TRIzol reagent (Invitrogen, Carlsbad, CA, USA). A 1 μg aliquot of RNA was converted to cDNA. The sequences of primers used for PCR amplification are listed in Table 3. qRT-PCR was performed using SYBR Green PCR master mix (Beyotime, Haimen, China) on stepone™ quantitative RT-PCR system (Applied Biosystems, Waltham, MA, USA). A 20 μL reaction mixture was amplified using the following thermal parameters: denaturation at 95 °C for 2 min, 40 cycles of the amplification step (95 °C for 15 s and 60 °C for 15 s), and a final extension at 95 °C for 15 s and 60 °C for 15 s. Glyceraldehyde 3-phosphate dehydrogenase (GAPDH) was served as the endogenous control. Relative mRNA expression levels were analyzed by the 2^−ΔΔCt^ method.

### 4.12. Statistical Analysis

All values were expressed as mean ± standard error of the mean (SEM). The difference between two groups was assessed using Student’s *t*-test, and multiple comparisons were performed by analysis of variance (ANOVA) followed by Tukey’s post hoc tests using GraphPad Prism 5 (GraphPad Prism Software Inc., San Diego, CA, USA). A value of *p* < 0.05 was considered to be statistically significant.

## Figures and Tables

**Figure 1 ijms-23-10465-f001:**
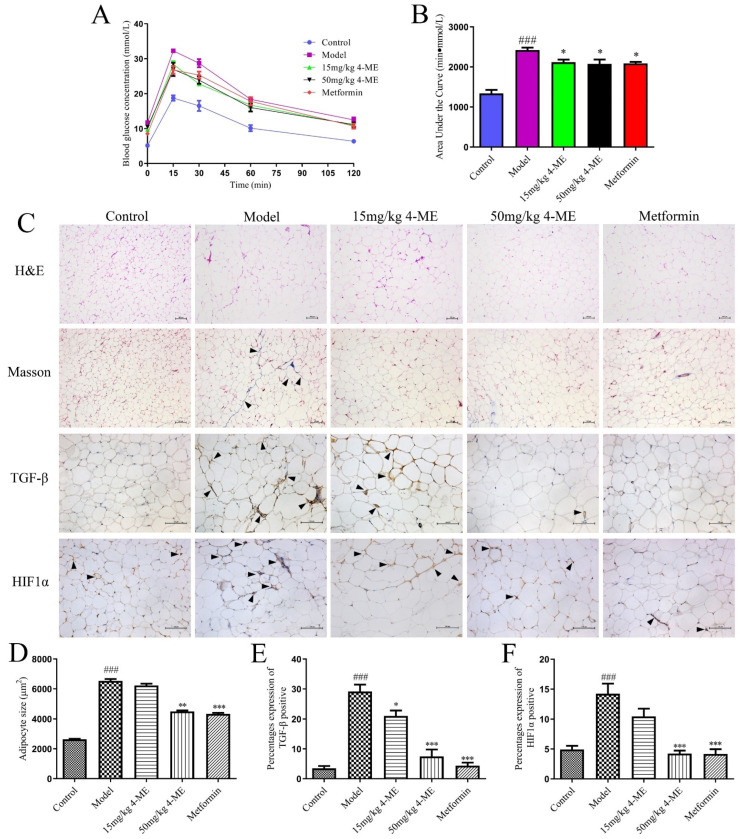
Effects of 4-ME upon intraperitoneal glucose tolerance test (IPGTT) and adipose tissue microenvironment in HFD-fed mice. (**A**) IPGTT (*n* = 5). (**B**) The area under the curve of IPGTT (*n* = 5). (**C**) The rows from the top to the bottom represent H&E staining (original magnification, 100×), Masson staining (100×), and immunostaining of TGF-β antibody (200×) and HIF1α (200×) in adipose tissue sections, respectively. (**D**) The size of adipocytes (*n* = 300). (**E**,**F**) Quantification for the percentage expression of TGF-β and HIF1α protein in panel (*n* = 4). Arrowheads indicate pathological changes. Values were given as mean ± SEM; ^###^
*p* < 0.001 compared with the Control group, * *p* < 0.05, ** *p* < 0.01, *** *p* < 0.001 compared with the Model group.

**Figure 2 ijms-23-10465-f002:**
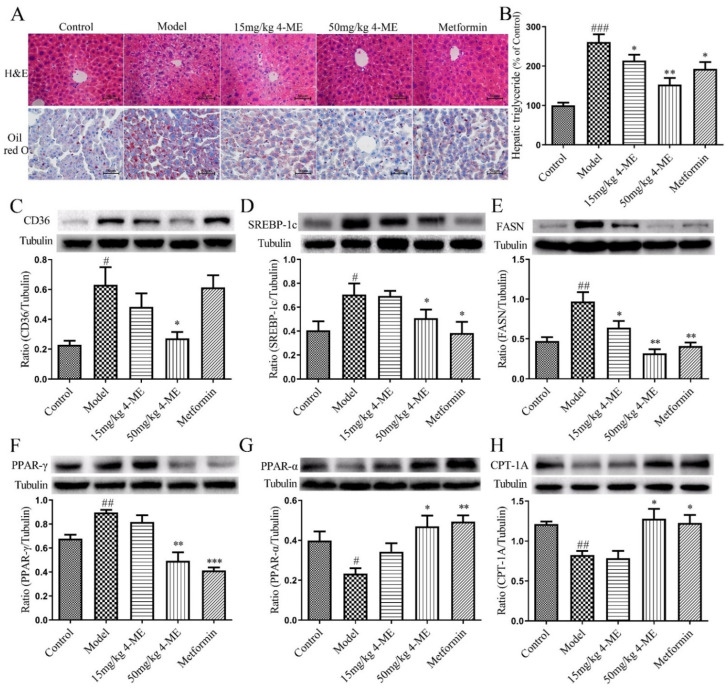
Effects of 4-ME on hepatic steatosis in HFD-fed mice. (**A**) Histological observation of the liver in experimental mice with H&E staining (original magnification, 400×) and Oil red O staining (400×). (**B**) Hepatic TG content (*n* = 8). (**C**–**H**) Shown are representative immunoblots and densitometric quantification of CD36, SREBP-1c, FASN, PPAR-γ, PPAR-α, and CPT-1A protein expressions (*n* = 3–4). The results were expressed as mean ± SEM; ^#^
*p* < 0.05, ^##^
*p* < 0.01, ^###^
*p* < 0.001 compared with the Control group, * *p* < 0.05, ** *p* < 0.01, *** *p* < 0.001 compared with the Model group.

**Figure 3 ijms-23-10465-f003:**
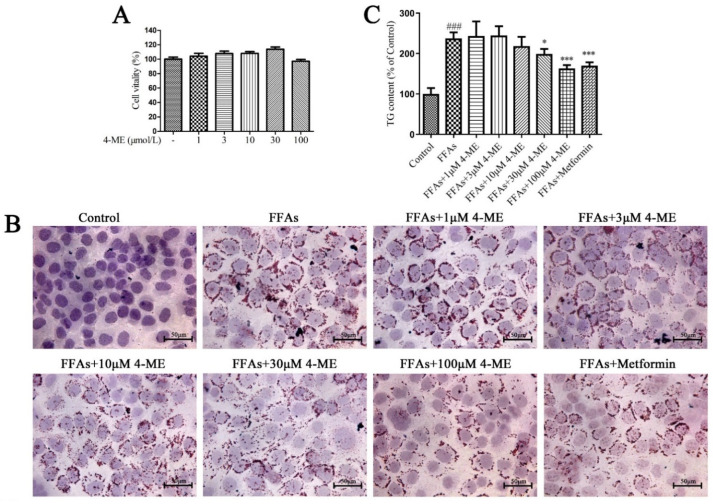
The 4-ME reduced lipid accumulation in FFA-induced LO2 cells. (**A**) Effects of different concentrations of 4-ME on cell viability (*n* = 8). (**B**) Effect of 4-ME on lipid accumulation was visualized using Oil red O staining. (**C**) Quantitative analysis of intracellular TG content (*n* = 6). (**D**–**I**) Effect of 4-ME on the expression of genes related to lipid metabolism in FFA-incubated LO2 cells (*n* = 3–4). The results were expressed as mean ± SEM; ^#^
*p* < 0.05, ^##^
*p* < 0.01, ^###^
*p* < 0.001 compared with the Control group, * *p* < 0.05, ** *p* < 0.01, *** *p* < 0.001 compared with the FFA group.

**Figure 4 ijms-23-10465-f004:**
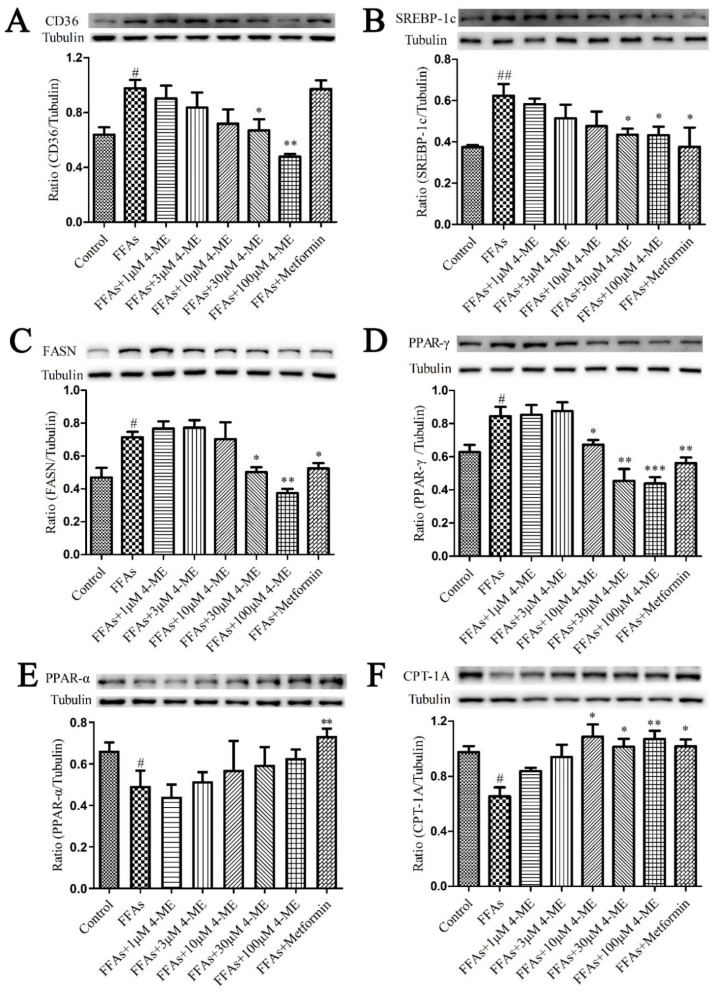
Effects of 4-ME on the expression of proteins involved in lipid metabolism in FFA-induced LO2 cells. Quantification data of CD36 (**A**), SREBP-1c (**B**), FASN (**C**), PPAR-γ (**D**), PPAR-α (**E**), and CPT-1A (**F**) protein expression levels are presented. The results were expressed as mean ± SEM (*n* = 3–4); ^#^
*p* < 0.05, ^##^
*p* < 0.01 compared with the Control group, * *p* < 0.05, ** *p* < 0.01, *** *p* < 0.001 compared with the Model group.

**Figure 5 ijms-23-10465-f005:**
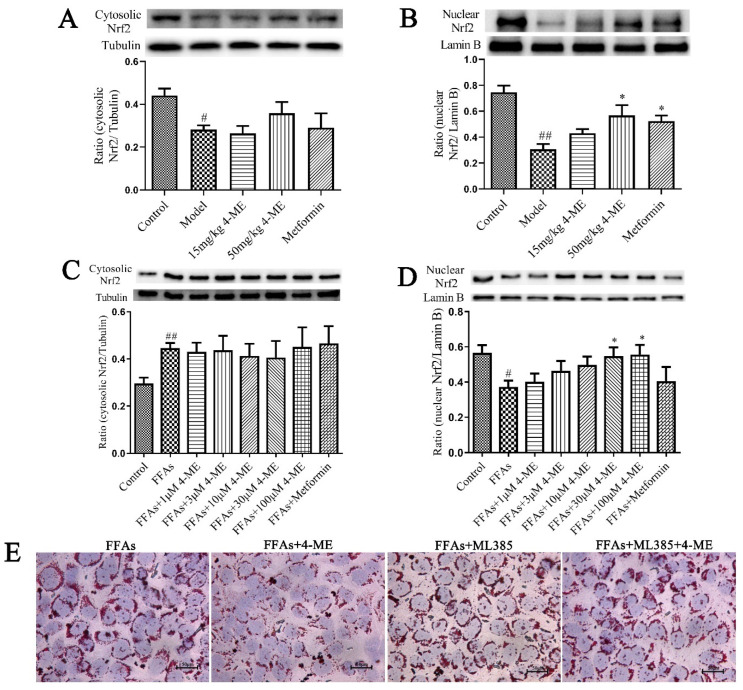
The 4-ME attenuated lipid metabolism by activating Nrf2. (**A**–**D**) The protein levels of cytosolic and nuclear Nfr2 in the liver or LO2 cells (*n* = 3–4). (**E**) Oil red O staining. (**F**) Intracellular TG content (*n* = 8). (**G**–**K**) Quantification data of Nfr2, CD36, SREBP-1c, FASN, and PPAR-γ protein expression levels in LO2 cells (*n* = 3–4). The results were expressed as mean ± SEM; ^#^
*p* < 0.05, ^##^
*p* < 0.01 compared with the Control group, * *p* < 0.05, ** *p* < 0.01, *** *p* < 0.001 compared with the Model group or FFA group, ^$^
*p* < 0.05, ^$$^
*p* < 0.01, ^$$$^
*p* < 0.001 compared with the FFAs+4-ME group.

**Figure 6 ijms-23-10465-f006:**
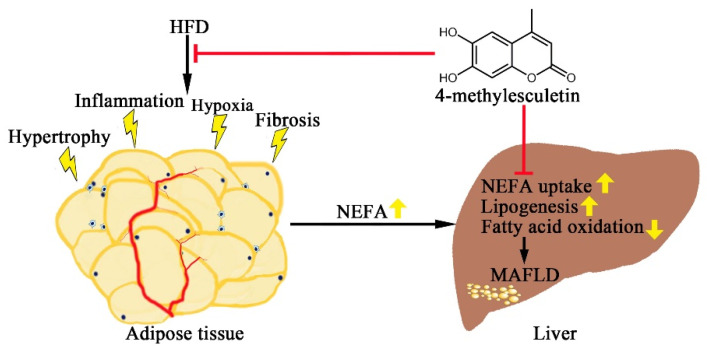
A depiction of effects of 4-ME upon obesity and MAFLD.

**Table 1 ijms-23-10465-t001:** Effects of 4-ME upon physiological parameters in HFD-fed mice.

Groups (*n* = 8)	Body Weight (g)					Epididymal Fat/Body Weight (mg/g)	The Lee Index
	0 Week	2 Week	4 Week	6 Week	8 Week		
Control	37.1 ± 0.4	36.4 ± 0.3	35.2 ± 0.8	35.5 ± 0.5	34.6 ± 0.97	15.6 ± 1.2	0.301 ± 0.001
Model	48.8 ± 2.0 ^###^	49.3 ± 2.3 ^###^	49.8 ± 2.9 ^###^	51.8 ± 3.1 ^###^	52.6 ± 3.1 ^###^	46.0 ± 6.8 ^###^	0.338 ± 0.003 ^###^
15 mg/kg 4-ME	48.8 ± 2.3	48.1 ± 2.0	48.9 ± 2.1	50.3 ± 2.2	49.5 ± 2.5	37.3 ± 2.8	0.331 ± 0.004
50 mg/kg 4-ME	48.6 ± 2.9	48.4 ± 2.6	48.0 ± 2.8	47.0 ± 1.1	45.8 ± 2.3 *	30.7 ± 2.8 *	0.320 ± 0.002 ***
Metformin	49.0 ± 2.0	48.1 ± 1.6	47.9 ± 1.4	46.7 ± 2.2	45.5 ± 1.1 *	29.3 ± 3.1 *	0.326 ± 0.003 *

^###^*p* < 0.001 compared with the Control group; * *p* < 0.05, *** *p* < 0.001 compared with the Model group.

**Table 2 ijms-23-10465-t002:** Effects of 4-ME on serum biochemical parameters in HFD-fed mice.

Groups (*n* = 8)	FBG (mmol/L)	Insulin (mIU/L)	HOMA-IR	Leptin (pg/mL)	HDL-C (mmol/L)	LDL-C (mmol/L)	T-CHO (mmol/L)	TG (mmol/L)	NEFA (μmol/L)
Control	4.9 ± 0.3	4.3 ± 0.3	0.91 ± 0.04	499.4 ± 31.2	1.2 ± 0.1	0.9 ± 0.1	3.7 ± 0.2	0.88 ± 0.03	441.1 ± 23.8
Model	11.7 ± 0.8 ^###^	6.1 ± 0.3 ^###^	3.17 ± 0.16 ^###^	634.5 ± 29.3 ^##^	0.9 ± 0.1	2.1 ± 0.2 ^###^	7.0 ± 0.6 ^###^	1.27 ± 0.12 ^##^	1023.0 ± 88.6 ^###^
15 mg/kg 4-ME	9.5 ± 0.5 *	5.8 ± 0.3	2.82 ± 0.34	610.2 ± 29.0	1.0 ± 0.1	2.1 ± 0.1	6.6 ± 0.4	1.25 ± 0.07	797.3 ± 121.1 *
50 mg/kg 4-ME	9.9 ± 0.6 *	5.3 ± 0.2 *	2.35 ± 0.12 **	548.8 ± 25.0 *	1.1 ± 0.2	1.7 ± 0.1	6.4 ± 0.5	1.13± 0.08	521.2 ± 18.9 ***
Metformin	8.7 ± 0.5 **	6.8 ± 0.2	2.53 ± 0.13 **	571.7 ±16.2	1.4 ± 0.2 *	1.2 ± 0.1 **	6.4 ± 0.3	1.18 ± 0.11	655.4 ± 48.8 **

^##^*p* < 0.01, ^###^
*p* < 0.001 compared with the Control group; * *p* < 0.05, ** *p* < 0.01, *** *p* < 0.001 compared with the Model group.

**Table 3 ijms-23-10465-t003:** Primer sequences for qRT-PCR.

Gene Name	Forward Primer (5′→3′)	Reverse Primer (5′→3′)
*CD36*	GAG AAC TGT TAT GGG GCT AT	TTC AAC TGG AGA GGC AAA GG
*SREBP1*	ACG GGA TGG ACT GAC TT	AGG CTT CTT TGC TGT GAG ATG
*FASN*	AAG GAC CTG TCT AGG TTT GAT GC	TGG CTT CAT AGG TGA CTT CCA
*PPARG*	ACT CCA CAT TAC GAA GAC AT	CTC CAC AGA CAC GAC ATT
*PPARA*	GCG AAC GAT TCG ACT CAA GC	TAC CGT TGT GTG ACA TCC CG
*CPT1A*	TGT CCA GCC AGA CGA AGA AC	ATC TTG CCG TGC TCA GTG AA
*GAPDH*	GTC TCC TCT GAC TTC AAC AGC G	ACC ACC CTG TTG CTG TAG CCA A

## Data Availability

Data will be made available by the corresponding author upon request.

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
