# Peer review of "Metabolic Syndrome Ameliorated by 4-Methylesculetin by Reducing Hepatic Lipid Accumulation"

_ijms, 2022, doi:10.3390/ijms231810465_

Round 1

Reviewer 1 Report

The present study evaluates the activity of 4-methylesculetin in a high-fat diet model of obesity. The authors report interesting findings related to the effects of the compound on adipose tissue pathology and liver fat accumulation. However, several notable concerns were noted.

Specific comments

1.       The authors did not present or discuss the mice's food intake during the study period. In a diet study like this one, it is difficult to interpret the findings in the absence of food intake data. If there was an effect on cumulative or daily food intake, then alternate mechanisms of 4-ME activity need to be examined.

2.       Body weight should be presented for each week of the study, if available. Presenting terminal body weights does not allow visualization of the compound effect that happens after the first 8 weeks of feeding.

3.       Figure 1C – the staining and the IHC images are not convincing on their own. Ideally, the authors should present a) high-quality images if available; b) image quantification, especially IHC quantification and adipose cell diameter if available/possible.

4.       Line 115-116: in text, the authors claim that TG and NEFA were decreased by 4-ME and metformin in a statistically significant manner. However, table 2 does not show statistical significance for the TG values.

5.       Figure 3 – it is unclear from the figure legend or the methods whether LO2 cells were pretreated/concurrently treated/post-treated with 4-ME with respect to FFA treatment. The duration of the treatments is also not mentioned. Please provide complete experimental details for all results

6.       What led to the selection of the 4-ME doses used in the animal and cell studies? Is there any bioavailability evidence for the compound at these doses in mice? The justification for the doses should be provided in the text.

7.       Discussion is very sparse. Authors need to provide additional context for their observations. A few issues that the authors can expand on are:

a.       The differences between 4-ME and Esculetin. There are several studies that have reported the liver effects of Esculetin including its property to reduce liver fat. (https://pubmed.ncbi.nlm.nih.gov/27769711/, https://pubmed.ncbi.nlm.nih.gov/28527877/, https://pubmed.ncbi.nlm.nih.gov/30263537/) Are there any structure-activity studies that explain the differences in the mechanisms of the liver fat accumulation between Esculetin and 4-ME? Is there any interconversion between Esculetin and 4-ME? Any prodrug-active drug relationship? Did the authors examine the mechanisms reported for Esculetin’s property of reduced liver fat (e.g., AMPK activation) when they studied the 4-ME effect?  This is important to further establish the novelty of the findings.

b.       While authors have identified nrf2 as a mediator or 4-ME effect in the liver, they haven’t reported a mechanism for 4-ME effect in the adipose. This is fine but it would be helpful to speculate on potential mechanisms leading to the effects of 4-ME in the adipose.

8.       A schematic/graphic depicting the proposed mechanism of 4-ME activity in the adipose tissue and the liver would be beneficial for the readers.

General comments:

1.       The manuscript needs English language review. The grammar and the word choice are less than ideal in several places including in the title. For example,

a.       in the title – I suggest replacing “through improving” with “by improving” and replacing “dampening” with “reducing”

b.       Line 63: the word “insight” is not the correct word choice here. I recommend “attention” instead of “insight”

Reviewer 2 Report

The manuscript by Li, et al. described the impact of one natural product, 4-Methylesculetin (4-ME), in a HFD-induced obesity model. By examining adipose tissue histology and liver lipid uptake and accumulation, authors reached a conclusion of beneficial effects of 4-ME in ameliorating metabolic syndrome. Overall, this study has scientific merits. However, I also identified several places to further strengthen the current paper. 

1. While the evidence of dampening hepatic lipid accumulation by 4-ME treatment is rather strong, the evidence of improvement in adipose tissue microenvironment is very weak. The histology data is insufficient to support the overall conclusion. Authors need to perform inflammatory marker (macrophage infiltration) staining, inflammatory gene expression analysis, and fibrosis gene expression analysis.

2. In addition to lipid accumulation, whether 4-ME has an anti-inflammatory and anti-fibrotic property also needs to be investigated.

3. I am curious the improvements in adipose tissue and in the liver are simultaneous events, or the improvement in adipose tissue premises benefits in the liver, or vice versa

Round 2

Reviewer 1 Report

Add a sentence when describing table 1 to indicate that no change in food intake was observed. Also include the graph as supplementary data.

Line 271: the word guaranteed should be replaced with warranted

Lines 390-391 in the statistical analysis section - ANOVA is repeated twice. 

Author Response

Point 1: Add a sentence when describing table 1 to indicate that no change in food intake was observed. Also include the graph as supplementary data.

Response:

Thanks for your precious comments and suggestions.

We have added the corresponding description in the revised version of the manuscript according to the comments and suggestions as follows.

Line 79-80: And 4-ME and metformin did not cause changes in food intake (Figure S1).

We have added the corresponding graph in the revised version of the supplementary data.

Point 2: Line 271: the word guaranteed should be replaced with warranted

Lines 390-391 in the statistical analysis section - ANOVA is repeated twice.

Response:

Thanks for your precious comments and suggestions.

We revised the manuscript according to the comments as follows.

Line 272: “guaranteed” à “warranted”

Lines 390-391: “multiple comparisons were performed by analysis of variance (ANOVA) followed by using One-way analysis of variance (ANOVA) by Tukey’s post hoc tests using GraphPad Prism 5” à “multiple comparisons were performed by analysis of variance (ANOVA) followed by Tukey’s post hoc tests using GraphPad Prism 5”

Reviewer 2 Report

The revision answered most of my questions. Since the focus is the liver, I would suggest remove "adipose tissue microenvironment" from the title. 

Author Response

The revision answered most of my questions. Since the focus is the liver, I would suggest remove "adipose tissue microenvironment" from the title.

Response:

Thanks for the precious comment and suggestion.

We revised the title in the manuscript according to the comments as follows.

“4-Methylesculetin Ameliorated Metabolic Syndrome by Improving Adipose Tissue Microenvironment and Reducing Hepatic Lipid Accumulation” à “4-Methylesculetin Ameliorated Metabolic Syndrome by Reducing Hepatic Lipid Accumulation”
